# Fluorinated-Triazole-Modified ZnO and Its Application in Marine Antifouling

Yu Yang, Guoqing Wang *, Longlin Lei, Yangkai Xiong, Zhiqiang Fang, Lei Huang, Jinbo Liu, Daxiong Hu and Jianhe Liao

State Key Laboratory of Marine Resource Utilization in South China Sea, School of Materials Science and Engineering, Hainan University, No. 58, Renmin Avenue, Haikou 570228, China; yangyu20210224@163.com (Y.Y.); 996033@hainanu.edu.cn (L.L.); x64880030@163.com (Y.X.); 15754332338@163.com (Z.F.); hl1842251332@163.com (L.H.); kimber000@163.com (J.L.); 18055437118@163.com (D.H.); 990359@hainanu.edu.cn (J.L.)
* Correspondence: wangguoqing@hainanu.edu.cn

**Abstract:** The accumulation of marine biological growth has irreversible negative effects on shipping and coastal fisheries. In this paper, a new antibacterial nanofiller—triazole fluoroaromatic hydrocarbon−modified nano−zinc oxide (ZnO−APTES−TRF)—was prepared by a Cu(I)−catalyzed azide–alkyne click chemical reaction. The modification of nano−ZnO with triazole ring fluoroaromatic hydrocarbons were testified by FT−IR, XPS, and EDS. The grafting rate of ZnO−APTES−TRF can reach 32.38%, which was verified by the TGA test. The ZnO−APTES−TRF was mixed with zinc acrylate resin to produce a low surface energy antifouling coating with a surface water contact angle of 106°. The bactericidal rate of ZnO−APTES−TRF against *Escherichia coli*, *Staphylococcus aureus*, and *Pseudoalteromonas* sp. can reach more than 98% due to the synergistic effect of triazole and fluorine. The 120−day marine experiment shows that the low surface energy antifouling coating of ZnO−APTES−TRF/ZA is expected to be widely used in the field of marine antifouling.

**Keywords:** antifouling; marine; ZnO; triazole; modification; hydrophobic

## 1. Introduction

The growth and accumulation of marine biological pollution on sea diving equipment has increased the volume and mass and adversely affected the operation of the hull, as well as increased the greenhouse gas emissions, consumed many natural as well as financial resources, and has adversely affected maritime shipping and coastal fishing [1,2]. Antifouling coatings were used early on as a long−term and effective protective measure to stem the spread of biological contamination and reduce its impact on underwater equipment [3,4].

Tributyltin (TBT) is widely used in the coatings industry for its broad−spectrum bactericidal property. However, the indiscriminate attack and teratogenicity of TBT on organisms kill the fouling organisms and destroy the diversity of the ecological environment in the ocean, and even accumulate in the human body [5,6]. Therefore, in 1999, the International Maritime Organization (IMO) passed an assembly resolution to impose a ban on the use of TBT by 1 January 2003 [7]. Cuprous is an effective biocide in coatings and is still widely used, but its effectiveness is not long−term when compared to TBT. The antifouling effect of heavy metal ions is superior but harms ecosystem circulation and resource use [8,9]. To protect the cleanliness of the surface of sea diving equipment and to not destroy the ecosystem, the development of environmentally friendly antifouling coatings is the task and topic of our time.

In this regard, organic/inorganic composite coatings have attracted much attention in the coatings industry. The combination of inorganic filler particles and organic resin can improve the mechanical strength of the layer and achieve a multifunctional antibacterial

effect by physicochemical means [10–15]. Among them, the application and research of nanofillers are the hotspots. Many studies have been done on the modification methods of nanofillers. Ivanova [16] et al. synthesized a nano−synthetic material named black silicon, whose surface morphology has a good inhibitory effect on both Gram−negative and Gram−positive bacteria. Sabzi [17] et al. modified nano−titanium dioxide (TiO$_2$) surface with aminopropyltrimethoxysiloxane to improve polyurethane coatings' dispersion and mechanical properties. Ji Young Ryu [18] et al. grafted a peptide−like polymer with the properties of decomposing proteins onto the TiO$_2$ surface by shrinkage polymerization, which can effectively decompose the protein and mucus adhered to the coating surface. Changquan Li [19] et al. grafted cetyltrimethoxysilane (HDTMS) onto the surface of nano−zinc oxide (ZnO) and then doped it into epoxy resin to form a superhydrophobic surface to realize antibacterial stain removal. In summary, the modification studies described have previously focused on the inherent defects of nanoparticles. Due to the large specific surface area and high surface energy, nanoparticles are readily adsorbed and agglomerated, which cannot easily be uniformly dispersed in the organic resin, resulting in coating surface defects.

Dimitrakellis [20] et al. pointed out that a low surface energy antifouling coating with new nano−antibacterial filler can achieve the effect of antibiological attachment and the dual development of an "active" antibacterial. In previous studies, the triazole ring structure widely used in biopharmaceutics can destroy the biomass membrane [21–27], which is usually combined with fluorine to achieve a synergistic antibacterial mechanism. The fluorine's strong electronegativity and hydrogen−like mimic effect [28–32] harm bacterial reproduction and growth. Nevertheless, the two structures are seldom involved in marine antifouling. Therefore, we selected nano−ZnO as the modified filler and modified its surface with a triazole ring and fluoroaromatic hydrocarbon so to obtain a lightweight nano−antibacterial filler as the "active" antibacterial. Furthermore, an antifouling coating was obtained after contained it in zinc acrylate resin. The low energy surface caused by fluorine can effectively prevent the adhesion of bacteria.

## 2. Experimental Section

### 2.1. Materials

Nano−zinc oxide (30 nm), triethylamine, 4−(trifluoromethyl) benzyl bromide, azide trimethylsilane (TMSA), tryptone, and yeast extract used in this study were purchased from (Shanghai, China) Macklin Biochemical Co., Ltd., as was the 3−aminopropyl triethoxysilane. Copper sulfate pentahydrate was manufactured by (Guangdong, China) Sci Tech Co., Ltd. Sodium ascorbate was manufactured by Xiya Chemical Technology (Shandong, China) Co., Ltd. The 3−bromopropyne was obtained from (Shanghai, China) Aladdin Biochemical Technology Co., Ltd. Anhydrous ethanol was purchased from Xilong Science. Zinc acrylate resin was made in the laboratory. The fluorescent live and dead cell stain LIVE/DEAD BacLightTM Bacterial Viability Kit L7012 used in the study was purchased from the Thermo Fisher Scientific website. The conductivity of ultra−pure deionized water used must not exceed 0.2 μs/cm@25 °C.

*Staphylococcus aureus*, *Pseudoalteromonas* sp., and *Escherichia coli* used in this experiment are from the State Key Laboratory of Hainan University.

### 2.2. Methods

#### 2.2.1. Preparation of ZnO−APTES

An amount of 6 g of ZnO particles (30 nm) were added to the solution in a 7:3 volume ratio of ethanol to deionized water, stirred at a magneton stirring speed of 200 rpm for 20 min at room temperature, and dispersed uniformly. Simultaneously, 0.8% by weight of 3−aminopropyltriethoxysilane (APTES) was added to a specified mass of deionized water and vibrated for 0.5 h in a CNC ultrasonic cleaner at 100% duty. Finally, APTES hydrolyzed by ultrasonic vibration was added to the uniformly dispersed ZnO suspension, and the reaction took place at room temperature at a speed of 800 rpm·min$^{-1}$ for 4.5 h. After the

response was complete, a 5 mL centrifuge tube was used to centrifuge at 8000 rpm·min$^{-1}$ for 10 min. The centrifuge product obtained was washed several times with ethanol and filtered and dried in a vacuum drying cabinet at 40 °C for 5 h. Finally, we received the APTES graft product called ZnO−APTES for later use.

### 2.2.2. Preparation of ZnO−APTES−F

In a 250 mL flask, 3 g of ZnO−APTES was weighed and added to a tetrahydrofuran solution and stirred at room temperature for 0.5 h at 200 rpm·min$^{-1}$. A 1 mol equivalent of 4−(trifluoromethyl) benzyl bromide (TBB) and a 1.1 mol equivalent of triethylamine was added and stirred at 75 °C for 2.5 h. After the solution was cooled, it was centrifuged, filtered, and then washed three times with tetrahydrofuran and dried in a vacuum oven at 40 °C for 4 h to obtain the reaction product ZnO−APTES−F.

### 2.2.3. Preparation of ZnO−APTES−TRF

A mixture of 0.01 mol CuSO$_4$·5H$_2$O, 0.02 mol sodium ascorbate (NaAce) and 1 mol 3−bromopropyne (PBE) was dropped into the stirred (200 rpm·min$^{-1}$ for a while) tetrahydrofuran (THF) solvent containing 3 g of ZnO−APTES. Then, the temperature was raised to 75 °C and kept for 5 h, with 1 mol of azide trimethylsilane (TMSA) added. Afterward, 1 mol TBB and a 1.1 mol equivalent of triethylamine (TEA) were added and reacted for 3 h. The product was washed in the THF/H$_2$O mixed solution and then washed three times in the ethanol/H$_2$O hybrid solution. Finally, the ZnO−APTES−TRF was obtained after being centrifuged, precipitated, and dried in a constant temperature vacuum oven at 40 °C for 6 h. The reaction scheme is shown in Figure 1.

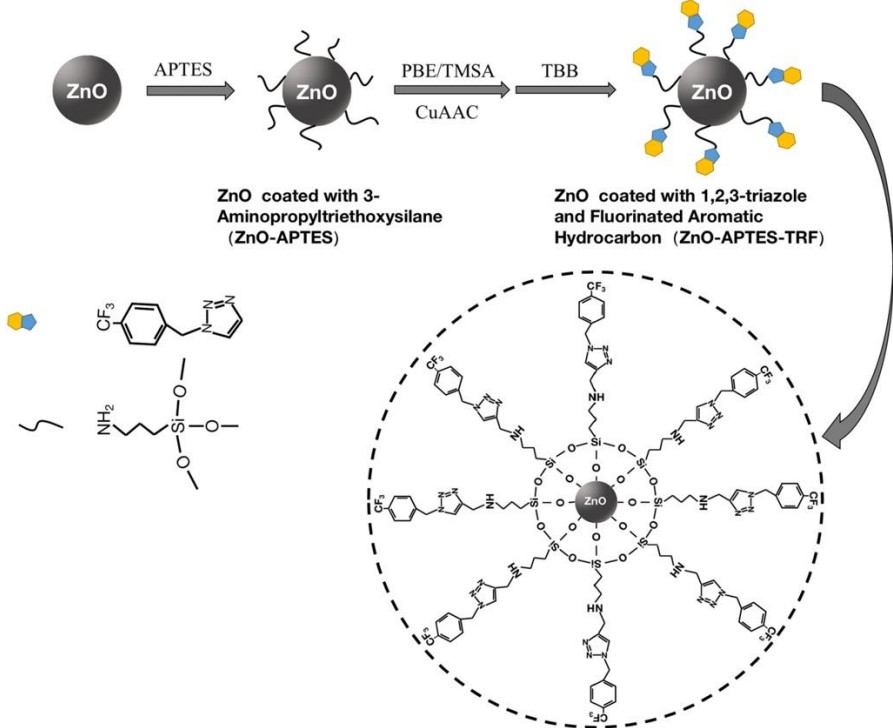

**Figure 1.** Modification and preparation process of ZnO−APTES−TRF.

### 2.2.4. Preparation of Composite Coatings

The composite coatings ZnO/ZA, ZnO−APTES/ZA, ZnO−APTES−F/ZA, and ZnO−APTES−TRF/ZA were prepared by mixing ZnO, ZnO−APTES, ZnO−APTES−F, and ZnO−APTES−TRF with self−made zinc acrylate resin (ZA), each with 10% mass fraction, respectively.

*2.3. Characterization*

2.3.1. Characterization of ZnO−APTES, ZnO−APTES−F and ZnO−APTES−TRF

The surface chemical structures of ZnO−APTES, ZnO−APTES−F, and ZnO−APTES−TRF were characterized by Fourier transform infrared spectroscopy (FT−IR−650(G)), and the wave number was 500–4000 cm$^{-1}$. The chemical elements on the surface of three different samples were analyzed by X−ray electron diffraction (XPS, Shimadzu Kratos, Axis Supra) with the Al target and a 0–1195 eV binding energy range. The graft content of the triazole ring fluoroaromatic hydrocarbons was detected by a thermogravimetric analyzer (TGA/DSC 3), in which the temperature range was RT−600 °C, and the heating rate was 10 °C·min$^{-1}$. Energy dispersive spectroscopy (EDS) characterized the surface element of the modified particles.

2.3.2. Antifouling Property Tests

The antibacterial properties of ZnO, ZnO−APTES, ZnO−APTES−F, and ZnO−APTES−TRF were characterized by two pioneering bacteria (*Escherichia coli* and *Staphylococcus aureus*) and *Pseudoalteromonas* sp., which is a common bacterium along the coast of China. All bacteria were cultured at 37 °C in a constant temperature bacterial culture shaker simulating seawater agitation at 120 rpm·min$^{-1}$ for 12 h to achieve 10$^8$ CFU/mL. We ran all tools used for testing in a 121 °C sterilization pot for 1.5 h. The sterilized test tools and specimens were placed under the UV lamp overnight. Samples of ZnO, ZnO−APTES, and ZnO−APTES−TRF dispersed in deionized water at a 0.1 μg/mL concentration were uniformly mixed with 10$^6$ CFU/mL bacterial solution, coated on the solid medium surface, and then incubated in a constant light temperature incubator for 24 h. Using the following Formula (1), calculate the sterilization efficiency:

$$M = \frac{MA - MB}{MA} \times 100\% \tag{1}$$

where *M* is the sterilization efficiency (%), *MA* is the number of colonies in the empty group, and *MB* is the number of colonies in the experimental group.

The antibacterial properties of ZA, ZnO/ZA, ZnO−APTES/ZA, ZnO−APTES−F/ZA, and ZnO−APTES−TRF/ZA composite coatings were evaluated by a fluorescence microscopic dead cell staining experiment: the prepared composite coatings were applied to 1 cm$^2$ glass plate, and then were placed in the culture medium with a fungal content of 10$^6$ CFU/mL and cultured in a constant temperature shaking table for 72 h in a simulated marine environment. At the end of the culture, we used fluorescent stain staining. Subsequently, they were observed and analyzed under the forward fluorescence microscope.

The water contact angle tested the hydrophobic state of the composite coating surface to examine the relationship between the hydrophobic angle and the antifouling performance of the coating.

To check the practicability of the ZnO−APTES−TRF/ZA composite coating in the natural marine environment, a pure ZA coating, ZnO/ZA coating, ZnO−APTES/ZA coating, ZnO−APTES−F/ZA coating, and ZnO−APTES−TRF/ZA coating were applied to the substrate in each case. At the Sinan−Kai, Xiuying District, Haikou City, Hainan Province, we took sea photos for 120 days and regularly recorded the plate surface adhesion.

## 3. Results and Discussion

*3.1. Characterization of ZnO−APTES, ZnO−APTES−F, and ZnO−APTES−TRF*

3.1.1. Microstructure Characterization

The infrared spectra of ZnO, ZnO−APTES, ZnO−APTES−F, and ZnO−APTES−TRF are shown in Figure 2. There is an infrared absorption peak of 1045.83 cm$^{-1}$ on the curve of ZnO−APTES, proving the existence of Si−O−Si, indicating that APTES successfully grafted onto the nano−ZnO. The characteristic peaks at 1327.76 cm$^{-1}$ and 1125.4 cm$^{-1}$ on the ZnO−APTES−F are the stretching vibration peaks of CF$_3$. The peak strength of C−N (1066.82 cm$^{-1}$) proves that fluoroaromatic hydrocarbons successfully modify the sur-

face of nanoparticles through a bromine substitution reaction with primary amine groups. Compared to the infrared spectra of ZnO−APTES−F, the peak intensities at 1553.87 cm$^{-1}$, 1460.56 cm$^{-1}$, 1223.18 cm$^{-1}$, 1052.13 cm$^{-1}$, and 791.92 cm$^{-1}$ on the ZnO−APTES−TRF curve proved the synthesis of the triazole ring group [33]. In addition, the absorption peaks at around 1125 cm$^{-1}$ and 1327 cm$^{-1}$ prove the existence of fluoroaromatic hydrocarbons. Therefore, we successfully modified the surface of the nano−ZnO with triazole ring fluoroaromatic hydrocarbons by a Cu (I)−catalyzed azide–alkyne cycloaddition click chemical reaction (CuAAC) [34–39].

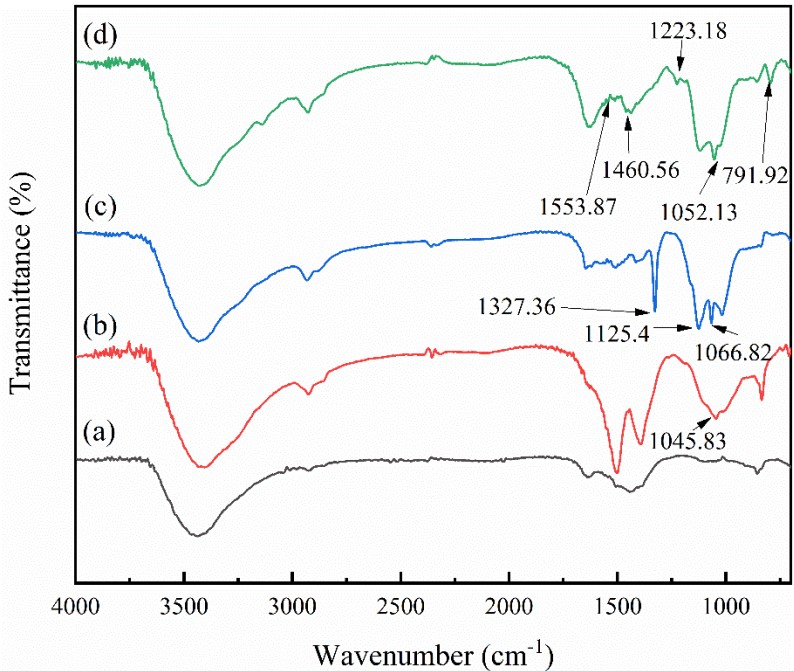

**Figure 2.** The FT−IR spectrum of (**a**) ZnO; (**b**) ZnO−APTES; (**c**) ZnO−APTES−F; (**d**) ZnO−APTES−TRF.

3.1.2. Surface Analysis

Figure 3a shows the XPS full spectrum curves of ZnO, ZnO−APTES, ZnO−APTES−F, and ZnO−APTES−TRF, respectively. It is obvious that the nitrogen peak appeared on the ZnO−APTES, ZnO−APTES−F, and ZnO−APTES−TRF curves, except for ZnO. Figure 3c–f show the peak curves of the N elements of the four samples. There is no signal peak of N on the surface of the ZnO. As can be seen from Figure 3d, C−N (397.9 eV) and −NH$_2$ (399.5 eV) prove that APTES successfully modified the ZnO surface. In Figure 3e, the peak intensity at 399.1 eV indicates the synthesis of C−NH−C, demonstrating that a substitution reaction occurs, transforming the primary amine group into a secondary amine. The three peaks located in Figure 3f: 397.9 eV, 398.3 eV, and 399.1 eV, represent the presence of C−N, −N=, and C−NH−C, respectively. These peaks prove the synthesis of the triazole ring on the ZnO surface. Figure 3b shows the fluorine peak diagrams of ZnO, ZnO−APTES, ZnO−APTES−F, and ZnO−APTES−TRF, respectively. The fluorine peaks are not present on the surfaces of the ZnO and ZnO−APTES, while they are on the ZnO−APTES−F and ZnO−APTES−TRF. This proves that the fluorinated aromatic hydrocarbons successfully modify the surface of the ZnO. Moreover, combined with the data in Figure 3f, we can conclude that the ZnO−APTES−TRF is a kind of successfully modifying ZnO, in which the triazole ring groups were wrapped up by fluorine−containing aromatic hydrocarbons, as shown in Figure 1.

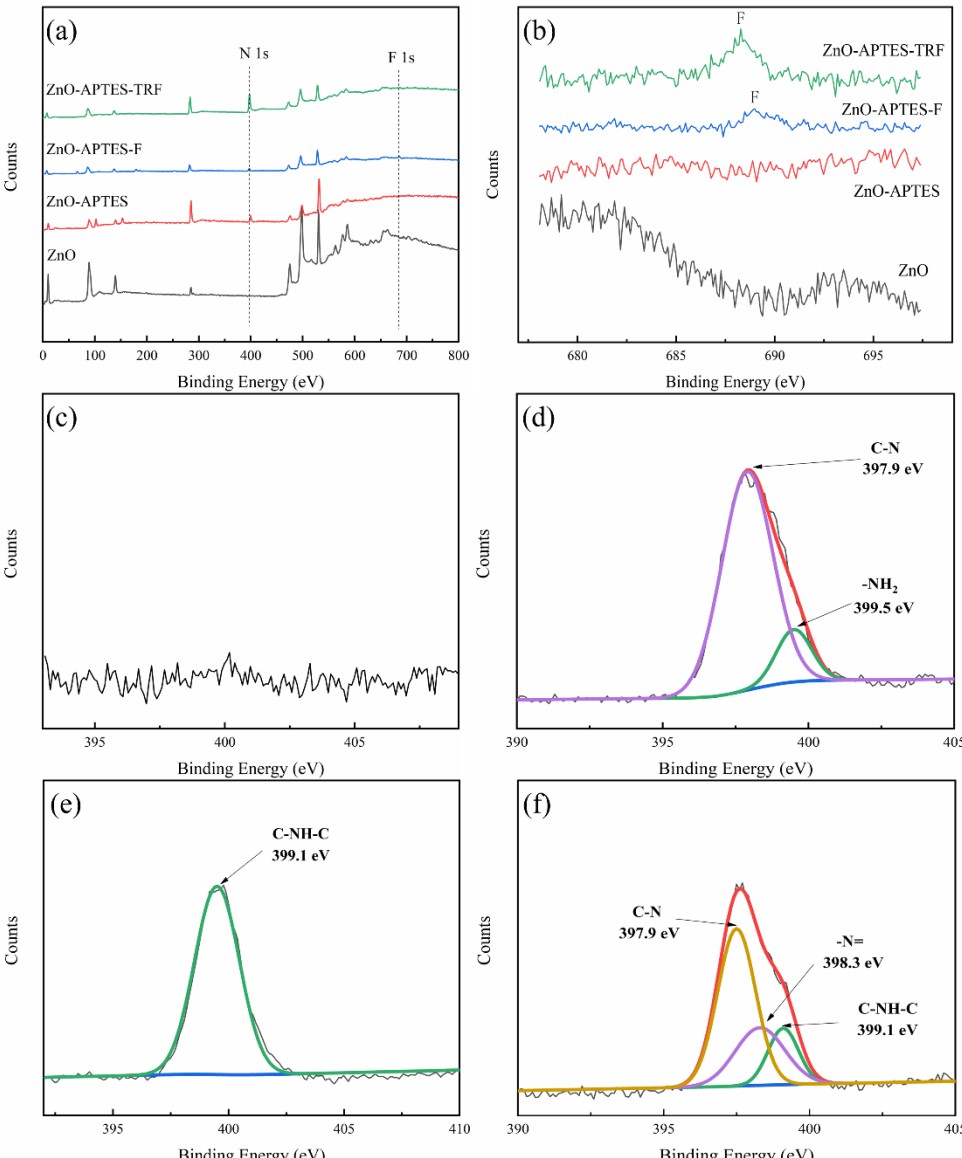

**Figure 3.** XPS spectrum analysis. (**a**) XPS spectrum; (**b**) XPS F1s peak diagram; (**c**) XPS N1s peak diagram of ZnO; (**d**) XPS N1s peak diagram of ZnO−APTES; (**e**) XPS N1s peak diagram of ZnO−APTES−F; (**f**) XPS N1s peak diagram of ZnO−APTES−TRF.

Figure 4 shows an EDS analysis of the surface elements of four samples. In Figure 4a,b, there is no fluorine on the surface of the ZnO and ZnO−APTES, while the fluorine content in Figure 4c,d is more than 2%. This proves that the fluorinated aromatic hydrocarbons successfully modified the surface of the ZnO. The strong electronegativity of fluorine will reduce the surface energy of the nanoparticles, thus increasing the hydrophobic effect of the coating. Moreover, the C, N, and F content is highest in Figure 4d, proving that there is a structural grafting with nitrogen and fluorine. Moreover, combined with the 13C NMR spectra in Figures S1 and S2, we can see that the ZnO surface was successfully modified by the triazole ring and trifluoromethyl aromatic hydrocarbon.

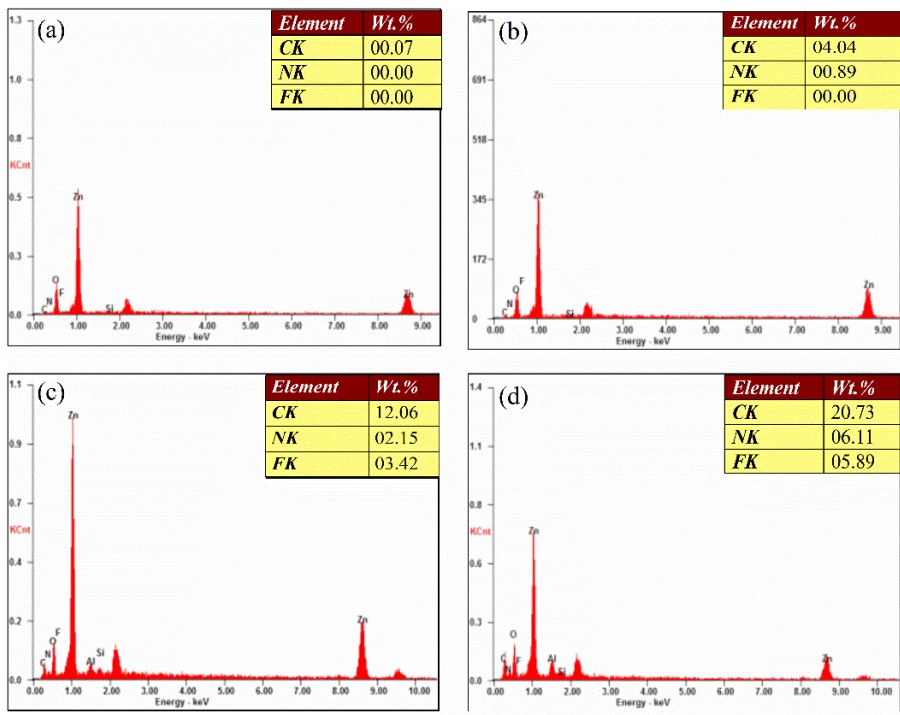

**Figure 4.** EDS energy spectrum analysis. (**a**) ZnO; (**b**) ZnO−APTES; (**c**) ZnO−APTES−F; (**d**) ZnO−APTES−TRF.

### 3.1.3. Graft Analysis

To verify the grafting content on the surface of modified particles, the TGA curve characterizing the heat loss weight of the ZnO−APTES, ZnO−APTES−F, and ZnO−APTES−TRF surfaces are shown in Figure 5. From the figure, the heat loss weight of the ZnO−APTES surface is about 8.47% compared to the ZnO, while the weight loss of the ZnO−APTES−F is about 29.97%, and the weight loss is about 20.10% compared to the ZnO−APTES. The above weight loss proves that the grafting content of fluoroaromatic hydrocarbons accounts for 20.10% of the whole system. Among them, the weight loss of the ZnO−APTES−TRF is up to 12.28% compared to the ZnO−APTES−F, proving that the grafting content of the triazole ring is 12.28% of the whole system.

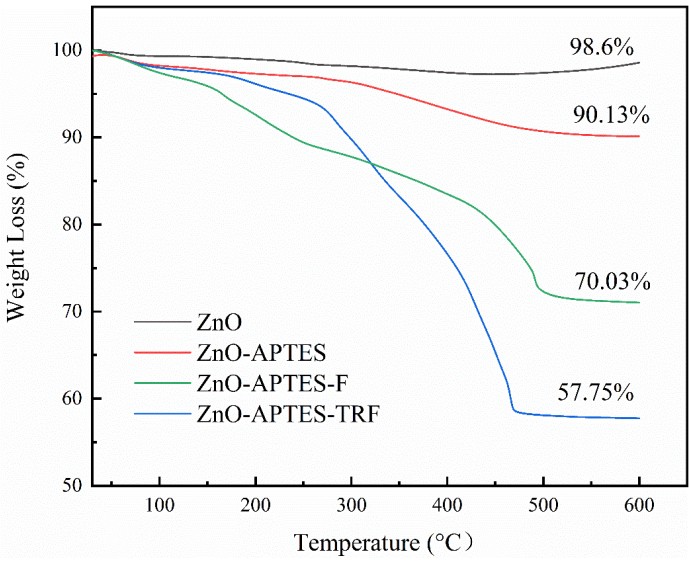

**Figure 5.** TGA curve analysis.

Thus, the weight ratio of the triazole ring and fluoroaromatic hydrocarbon within the ZnO−APTES−TRF can be calculated by the Formula (2):

$$MR = \left| \frac{NT - NF}{NF - NA} \right| \tag{2}$$

where *MR* represents the estimated weight ratio of triazole ring to fluoroaromatic hydrocarbon, *NT* is the weight loss of the ZnO−APTES−TRF, *NF* is the weight loss of the ZnO−APTES−F, and *NA* is the weight loss of the ZnO−APTES. According to the weight loss, the calculation results of *MR* are about 1:2, which is consistent with the theoretical value based on the ratio of our original substance. The actual proportion of the grafted functional groups is close to the theoretical value, which proves that the product is grafted according to the designed proportion. In summary, it demonstrates that the surface of the nano−ZnO is successfully grafted by the triazole ring fluoroaromatic hydrocarbons.

Therefore, the grafting content of the triazole cyclofluoroaromatic hydrocarbons on the surface of the ZnO−APTES−TRF can be concluded by Formula (3), which can reach 32.38%:

$$ML = |NT - NA| \tag{3}$$

where *ML* is the weight proportion of the triazole ring fluorinated aromatic hydrocarbons (%).

### 3.2. Antibacterial Properties of ZnO−APTES−TRF

The above characterization methods prove that TRF was successfully grafted onto the surface of the nano−ZnO. To verify the antibacterial activity of the modified powder, we used the bacterial plate counting method to count the number of surface colonies and recorded the results. The antibacterial plate count experiment of the three bacteria is shown in Figure 6a. Nano−ZnO has a weak inhibitory effect on the three bacteria due to the photocatalytic antibacterial properties of the nano−ZnO itself. After the modification of APTES, the antibacterial effect of the nano−ZnO increases, and the number of colonies on the solid medium decreases relatively. Due to the transformation of APTES, the dispersing performance of nanoparticles is successfully improved, increasing the contact area between the nano−ZnO and bacteria and the antibacterial effectiveness. The number of bacteria on the ZnO−APTES−F solid medium decreased, which may be due to the specific electronegativity of fluorine, which interferes with the continuous growth of colonies. There was almost no colony growth on the ZnO−APTES−TRF solid medium, proving the good antibacterial performance of the triazole ring medium.

The colony count histogram in Figure 6b is obtained by area counting. It is obvious that in the ZnO−APTES−TRF solid medium, the number of colonies of the three bacteria is the least, proving that the antibacterial performance of ZnO is successfully optimized after triazole ring−grafting. Moreover, according to the sterilization rate calculation in Figure 6c, the lethal rate of the ZnO−APTES−TRF for the three kinds of bacteria is more than 98%, which is far higher than the other three kinds of powders. This phenomenon successfully confirmed that the antifouling performance of ZnO is improved through the synergistic antibacterial mechanism of the triazole ring and fluoroaromatic hydrocarbons.

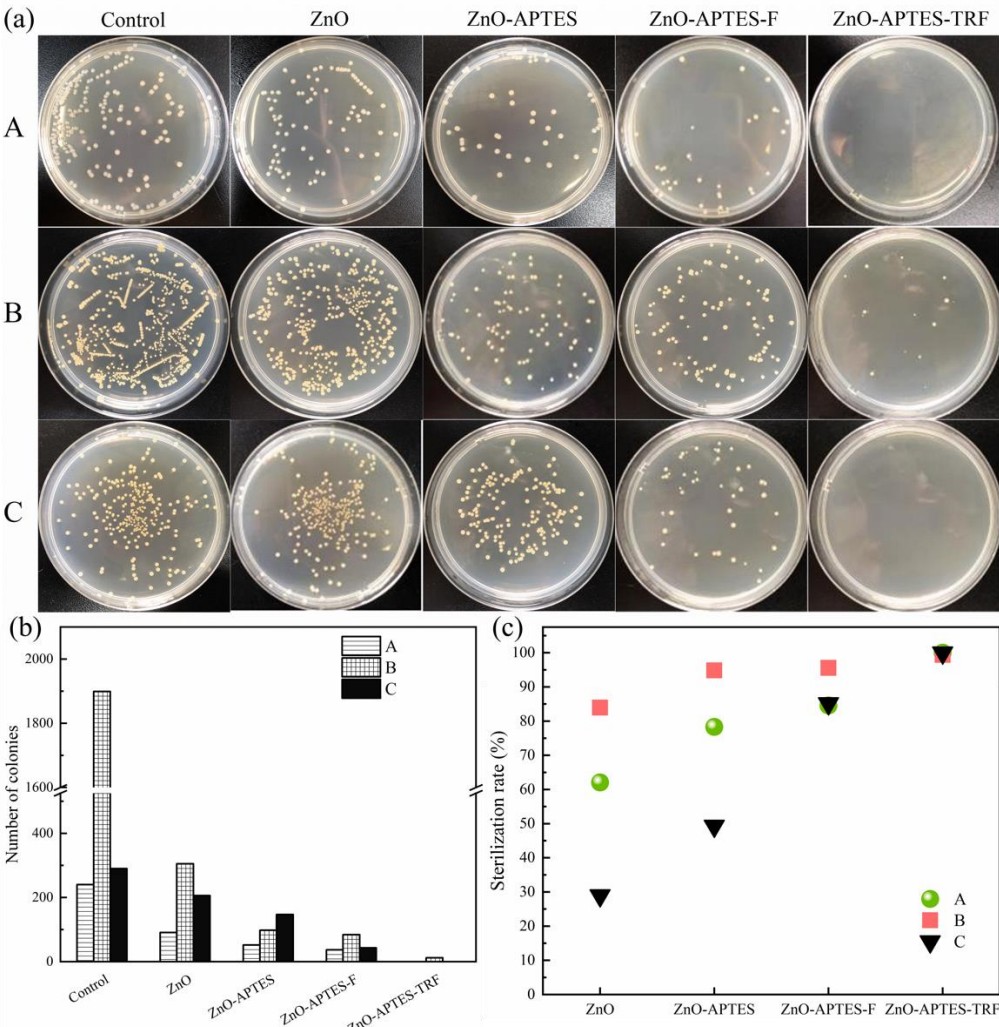

**Figure 6.** (**a**) Plate counting experiment of three kinds of bacteria. A: *Escherichia coli*; B: *Staphylococcus aureus*; C: *Pseudoalteromonas* sp.; (**b**) Column chart of colony number; (**c**) Sterilization rate.

### 3.3. Antibacterial Property Test of Coating

To study the antibacterial properties of the ZnO−APTES−TRF in antifouling coatings, we performed 48 h bacterial culture fluorescence staining experiments on the pure ZA (C0) coating, ZnO/ZA (C1) coating, ZnO−APTES (C2) coating, ZnO−APTES−F (C3) coating, and ZnO−APTES−TRF(C4) coating, respectively. The experimental results of the fluorescent staining of dead bacteria are shown in Figure 7a. According to the microscopic observation, the fluorescent area of the deadly red bacteria on the coating surface gradually increases with the different nano−ZnO powder additions, which is consistent with the results in Figure 6. In Figure 7a, the number of red dead cell spots on the surface of the C0 coating is the least, which is attributed to the specific antibacterial properties of the zinc acrylate resin, but the effect is poor. The red fluorescent spots on the surface of the C1 coating added with ZnO increased slightly, but less than that of the C2 layer added with the ZnO−APTES. The number of red fluorescent spots is the most on the surface of the C4 layer, proving that the zinc acrylate resin added with the ZnO−APTES−TRF has an excellent antibacterial effect. The experimental results in Figure S3 also prove that the ZA coating containing the ZnO−APTES−TRF has excellent antibacterial properties.

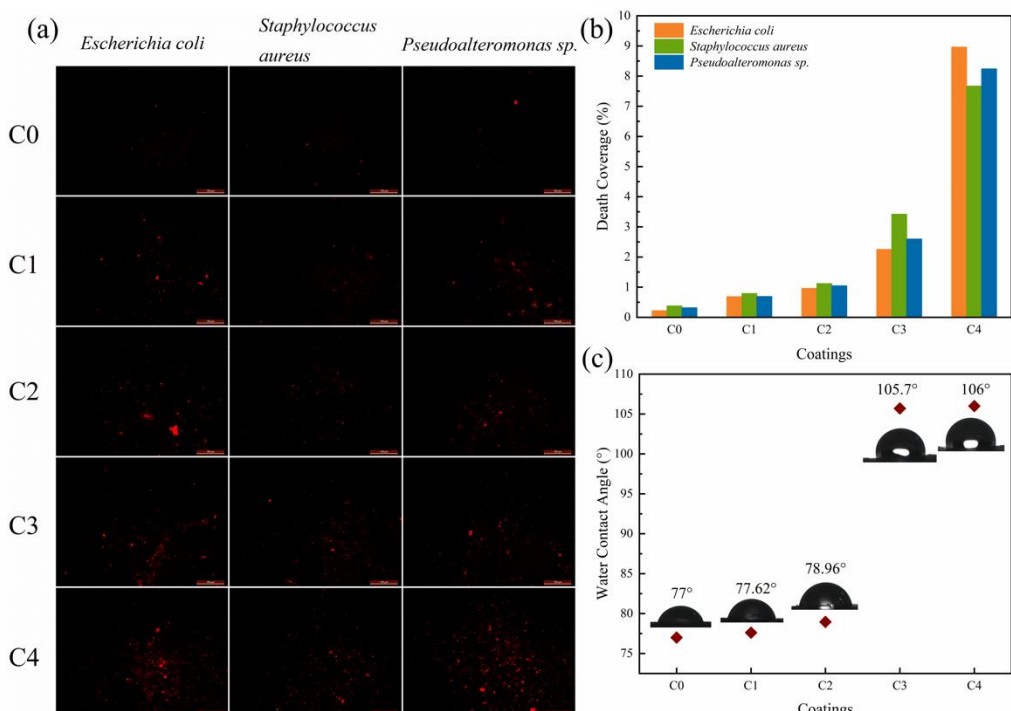

**Figure 7.** (**a**) Fluorescence staining micrograph of dead cells of *Escherichia coli*, *Staphylococcus aureus*, and *Pseudoalteromonas* sp.; (**b**) Dead cell coverage ratio (%); (**c**) Water contact angle value. (C0: Pure ZA coating; C1: ZnO/ZA coating; C2: ZnO−APTES/ZA coating; C3: ZnO−APTES−F/ZA coating; C4: ZnO−APTES−TRF/ZA coating).

The histogram in Figure 7b shows the dead bacteria area coverage of five different coatings on *Escherichia coli*, *Staphylococcus aureus*, and *Pseudoalteromonas* sp. The number of dead bacteria coverage areas of the pure ZA coating is the smallest, and the finished bacteria coverage area of the three bacteria is less than 0.5%. When the pure nano−ZnO is added, the dead bacteria coverage area on the coating surface increases, which is attributed to the bactericidal effect of the ZnO. The dead bacteria coverage area of the C2 coating is more extensive than that of the C1 layer, which is due to the surface modification of the nano−ZnO by the silane coupling agents that improve the dispersing performance of the nano−ZnO in the coating and enhances the antibacterial effect of the nano−ZnO. When the surface of the ZnO is modified with fluoroaromatic hydrocarbons, the dead bacteria covered area of the three bacteria can exceed 2%, proving that the grafting of fluoroaromatic hydrocarbons can further improve the antibacterial effectiveness of the ZnO. The dead coverage area of the three bacteria on the surface of C4 is more than 7%, which means that the presence of a triazole ring group further improves the antibacterial effectiveness based on fluorine−containing aromatic hydrocarbons.

Figure 7c shows the scatter diagram of the water contact angle of five coating surfaces. According to the figure, the addition of the ZnO−APTES−TRF and ZnO−APTES−F improves the hydrophobic performance of the coating surface, and the maximum water contact angle reaches 106° due to the low surface energy of the trifluoromethyl covered on the surface. In combination with Figure 7b,c, the bacteriostatic effect of the surface gradually increases with the increasing water contact angle.

In conclusion, the results demonstrate that the presence of fluorine element and triazole ring plays a positive role in the antibacterial effect of the coating.

It is reasonable to speculate that the good antibacterial performance of the ZnO−APTES−TRF/ZA coating is due to the synergistic effect of the triazole ring and fluoroaromatic hydrocarbons. On the one hand, the triazole ring can destroy the biomass membrane by inhibiting the synthesis of ergosterol and fundamentally preventing the wanton production

of fungi. On the other hand, through the lipophilicity and simulation performance, fluorine atoms act as a hook to penetrate the biomass membrane of the fouling organisms. Moreover, the strong electronegativity of the fluorine hurts the genetic material of the marine fouling organisms when it enters the biomass membrane through a hydrogen−like simulation effect, resulting in the abnormal development of the genetics of marine fouling organisms or the prevention of reproduction. Such an internal and external dual defense mechanism significantly improves the antifouling efficiency of the nano−ZnO.

### 3.4. Marine Environment Test

Two types of antibacterial laboratory tests show that the modified nanofiller has excellent antibacterial properties but cannot simulate the complex marine ecology due to the limitations of laboratory conditions. Therefore, the long−term antifouling performance of the C0, C1, C2, C3, and C4 composite coatings was evaluated by a marine environment test. As shown in Figure 8, the panel in the figure was immersed in the sea for 120 days. There are apparent adhesions of barnacles, algae, mussels, and other hard and soft organisms on the surface of the C0 coating, indicating that there is severe biological pollution in the port's marine environment. C1, C2, and C3 characters also have evident biofouling adhesion. However, the C4 coating surface containing the ZnO−APTES−TRF shows no significant biofouling, except for slight sediment and an organic biomass film.

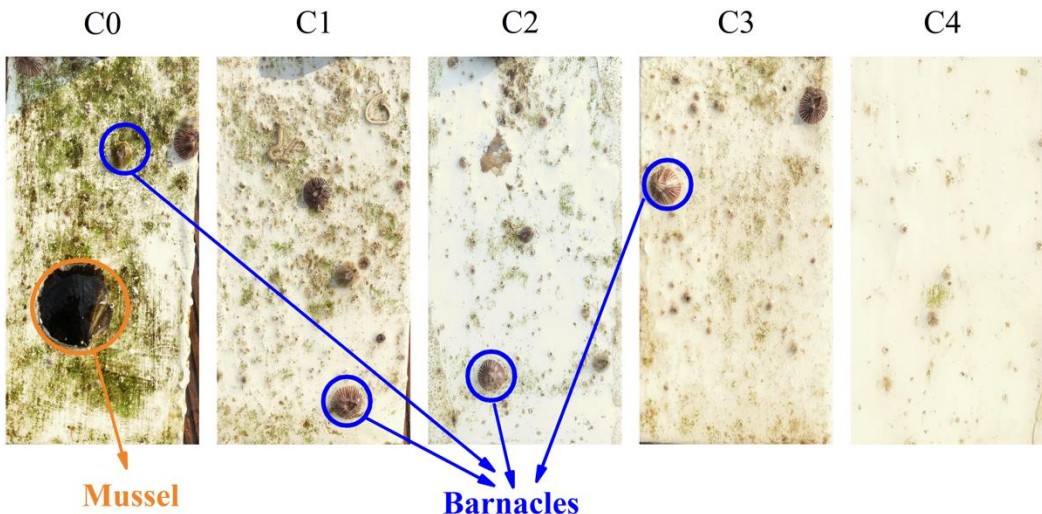

**Figure 8.** Coating surface fouling and adhesion after immersion for 120 days. (C0: ZA; C1: ZnO/ZA; C2: ZnO−APTES/ZA; C3: ZnO−APTES−F/ZA; C4: ZnO−APTES−TRF/ZA).

The complicated organisms represented by the barnacles can adhere and grow on the surface of the coating. Barnacles can be attached to the substrate surface by making barnacle glue [40,41]. When barnacle glue can no longer adhere smoothly to the coating surface, barnacle larvae fall off and seek the nearest suitable habitat. The low surface energy coating surface can be self−cleaning under the sheer power of seawater, and then make part of the larvae fall off quickly. Therefore, it can effectively prevent the growth and attachment of barnacles. In addition, the adhesion of some fungi penetrates the coating surface through a mucus secreted by themselves. Although the low surface energy coating can achieve the surface self−cleaning effect with the help of the sea current, its hydrophobicity does not reach the super sliding effect. Therefore, influential antibacterial functional groups are required for essential antibacterial agents. The coating surfaces of C3 and C4 can achieve this effect. On the C3 coating surface, there is little adhesion of barnacles and algae, while on the C4 surface, there is no growth of other problematic organisms, except for the bare filler caused by the ZA resin self−polishing.

The above phenomena show that the grafted ZnO−APTES−TRF modified filler still has an excellent antibacterial effect in the complex marine environment. The synergistic antifouling of fluorine and triazole ring on the surface of the ZnO−APTES−TRF destroys the soft material growth and indirectly prevents the growth of algae and mussels on the coating. Therefore, the C4 layer can maintain the antifouling effect for up to 120 days in the actual marine environment and has good long−term use. In addition, the wide application of the triazole ring and fluorine−containing aromatic hydrocarbons in biopharmaceuticals has proved their low toxicity and high safety. Their use in modified fillers is consistent with today's environmental concerns, which have potential application value in marine fisheries, aquaculture, shipping, and other industries.

## 4. Conclusions

In this study, we synthesized a novel antibacterial nanofiller, ZnO−APTES−TRF, by the CuAAC click reaction and modified the surface of the nano−ZnO with triazole ring fluoroaromatic hydrocarbons, which were testified by FT−IR, XPS, and EDS. The antibacterial mechanism of the ZnO−APTES−TRF was proposed, which is the synergistic effect of triazole ring and fluoroaromatic hydrocarbons. The main conclusions are:

(1) The antibacterial efficiency of fluorinated aromatic hydrocarbons and triazole rings was studied by antibacterial laboratory tests, which showed that fluorinated aromatic hydrocarbons enhanced the antibacterial efficiency of ZnO, and the presence of the triazole ring group greatly enhanced the antibacterial activity of the nano−ZnO. The inhibition rate of the ZnO−APTES−TRF on *Escherichia coli*, *Staphylococcus aureus*, and *Pseudoalteromonas* sp. can reach more than 98%. The ZnO−APTES−TRF/ZA coating can keep the antifouling effect in the natural marine environment for more than 120 days and has good long−term use.

(2) The rate of grafting of triazole ring fluorinated aromatic hydrocarbons on the surface of the ZnO−APTES−TRF can reach 32.38%. Through verification, the weight ratio of triazole ring to fluoroaromatic hydrocarbon is close to the theoretical value of 1:2, which is in line with the expectation. When coated with the ZA resin, the water contact angle of the ZnO−APTES−TRF/ZA surface can reach 106°, indicating a hydrophobic surface.

(3) Due to the wide application of triazole ring groups and fluorinated aromatic hydrocarbons in biomedicine, the ZnO−APTES−TRF with a low toxicity and high safety has potential application value in marine fisheries, aquaculture, shipping, and other industries.

**Supplementary Materials:** The following supporting information can be downloaded at: https://www.mdpi.com/article/10.3390/coatings12060855/s1, Figure S1: Liquid nuclear magnetic resonance carbon spectrum (13C NMR) of 1,2,3-triazole cyclohexane fluoroaromatic hydrocarbons (TRF); Figure S2: Solid state nuclear magnetic resonance carbon spectrum (13C NMR) of ZnO-APTES-TRF; Figure S3: Re-experimental diagram of fluorescent staining of dead cells of composite coating. (C0: Pure ZA coating; C1: ZnO/ZA coating; C2: ZnO-APTES/ZA coating; C3: ZnO-APTES-F/ZA coating; C4: ZnO-APTES-TRF/ZA coating).

**Author Contributions:** Conceptualization, G.W. and Y.Y.; methodology, Y.Y.; validation, Y.Y., Z.F. and L.H.; formal analysis, Y.Y. and Y.X.; investigation, L.L.; resources, G.W.; data curation, Y.Y. and J.L. (Jinbo Liu); writing—original draft preparation, Y.Y. and L.L.; writing—review and editing, G.W.; visualization, D.H.; supervision, G.W.; project administration, G.W. and J.L. (Jiaohe Liao); funding acquisition, G.W. All authors have read and agreed to the published version of the manuscript.

**Funding:** This research was funded by the National Natural Science Foundation of China (grant number 51963008) and Zhanjiang Baosteel New Building Materials Technology Co., Ltd. (grant number HD−KYH−2020105).

**Institutional Review Board Statement:** Not applicable.

**Informed Consent Statement:** Not applicable.

**Data Availability Statement:** Data sharing is not applicable to this article.

**Acknowledgments:** I would like to thank the support from the Analysis and Testing Center of Hainan University.

**Conflicts of Interest:** The authors declare no conflict of interest.

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
