# Peer review of "Fluorinated-Triazole-Modified ZnO and Its Application in Marine Antifouling"

_coatings, doi:10.3390/coatings12060855_

Round 1
Reviewer 1 Report
The paper is interesting and presents a method to prepare antibacterial coatings. Before it can be published it requires some modifications/corrections.
1. Literature. Several methods to prepare antibacterial surfaces already exist. Authors should cite more examples about different approaches in the introduction section. Examples: mechanical bactericidal effect, antibacterial action through wetting control or antibacterial action through combination of wetting control and bactericidal agent. Some examples including review papers:
Bactericidal Action of Smooth and Plasma Micro‐Nanotextured Polymeric Surfaces with Varying Wettability, Enhanced by Incorporation of a Biocidal Agent, P Dimitrakellis, K Ellinas, GD Kaprou, DC Mastellos, A Tserepi, Macromolecular Materials and Engineering 306 (4), 2000694
The papers by Ivanova et al.
i.e. Nature Communications, volume 4, Article number: 2838 (2013)
etc.
2. The resolution of the figures should be improved
3. I don't fully agree with the last statement
Due to the wide application of triazole ring groups and fluorinated aromatic hydrocarbons in biomedicine, ZnO-APTES-TRF has eco-friendly properties. This new anti-bacterial nanofiller with low toxicity and high safety has potential application value in marine fisheries, aquaculture, shipping, and other industries.
Fluorine containing coatings can not be easily considered as environmental friendly. Please rephrase.
Author Response
We have answered your questions in the uploaded file.

Reviewer 2 Report
This manuscript submitted by Yu Yang describes the anti-fouling performance of the fluoro-triazole modified nano ZnO and its application in zinc acrylate resin. The structure of the synthesized compounds was confirmed by FTIR spectroscopic technique. The experiments were performed with good logic and the results is worth to be published. However, there are some points to overcome for acceptance. I recommend the acceptance of this manuscript after a major revision followed by the editorial correction.
1. The title of the manuscript does not sound scientific, I suggest authors modify it so that it can reflect the scientific contribution of the research work.
2. Add some key findings in the abstract section.
3. In the introduction section, authors should clear the air of the research following with the previous work and the advancement of the current work.
4. To the confirmation of the synthesized compounds authors can perform 1H NMR.
5. All the figures need to modify in high resolution (300 dpi).
6. Too small replication: According to M&M only the Antibacterial property test of coating experiment was performed only once. It is recommendable to perform twice with 3 replications at least.
7. Statistical analysis with pair-wise testing using the Student’s t-test need to be performed, and significant changes indicate by asterisks and also need to state what error bars on graphs represent (sd/se/CI?). I would also suggest plotting individual data points on graphs instead of/alongside the means as this increases transparency and gives the reader a clearer idea of whether the assumptions made by statistical analysis. Additionally, you could use a Kruskal-Wallis test followed by Mann-Whitney tests adjusted for multiple comparisons to compare with untreated control.
8. Throughout the paper the bacterial species names need to be italicized.
9. Pseudoalteromonas is a genus name author need to specify which strain of Pseudoalteromonas was used in this study?
10. Unwanted spacing and typo mistakes throughout the manuscript. Need to be checked and correct carefully.
11. Too less discussion about the results. More discussion is needed.
12. Double check the way of adding references in the main text body. It looks different from the guide of this journal. Additionally, the reference formats do not match to this journal.
Author Response

(The authors gave the same response as above.)

Reviewer 3 Report
Author conducted basic study and characterization of the nano-material for anti fouling performance. Author needs more attention to the prepared manuscript that shows the lack of concentration. The quality of presentation is not good enough. Picture quality is worst, non- uniformity of fonts and it needs extensive editing before submission.
- Abstract: Please add quantitative results and provide appropriate reasoning. It is lengthy and reduce the abstract size.
- Try to reduce acronyms from the abstract.
- In introduction section, focus of the work is not clear, explain well.
- [Experimental section] The remarks information for chemicals and testing equipments in this work should be given, for example ( Model, Name of supplier, City, Country).
- Found few spell mistakes in the current version. Avoid the typo errors and unnecessary space in the written manuscript.
Author Response
Response to Reviewer #3 Comments
coatings-1758988: “Fluorinated-triazole modified ZnO and its application in marine antifouling” by Yang et al.
(Note: In the following the reviewer’ comments are in black, and our responses are in red. The modifications have been marked in highlight in the resubmitted manuscript.)
Author conducted basic study and characterization of the nano-material for anti fouling performance. Author needs more attention to the prepared manuscript that shows the lack of concentration. The quality of presentation is not good enough. Picture quality is worst, non-uniformity of fonts and it needs extensive editing before submission.
Thank you for your careful review and the constructive comments that help to upgrade the quality of our manuscript.
- Abstract: Please add quantitative results and provide appropriate reasoning. It is lengthy and reduce the abstract size.
Response1: Thanks for your kind reminder. Some key findings were added in the abstract which is shown as following:
Accumulation of marine biological growth have irreversible negative effects on shipping and coastal fisheries. In this paper, a new antibacterial nanofiller—triazole fuoroaromatic hydrocar-bons modified nano zinc oxide (ZnO-APTES-TRF)—was prepared by a Cu(I)-catalyzed az-ide-alkyne clicks chemical reaction. The modification nano ZnO with triazole ring fluoroaro-matic hydrocarbons were testified by FT-IR, XPS, and EDS. The grafting rate of ZnO-APTES-TRF can reach 32.38% which was verified by the TGA test. The ZnO-APTES-TRF was mixed with zinc acrylate resin to produce a low surface energy antifouling coating with a surface water contact angle of 106°. The bactericidal rate of ZnO-APTES-TRF against Escherichia coli, Staphylococcus aureus, and Pseudoalteromonas sp. can reach more than 98% due to the synergistic effect of triazole and fluorine. The 120-day marine experiment shows that the low surface energy antifouling coating ZnO-APTES-TRF/ZA is expected to be widely used in the field of marine antifouling.
- Try to reduce acronyms from the abstract.
Response2: Unnecessary acronyms in the abstract have been removed.
- In introduction section, focus of the work is not clear, explain well.
Response3: Thank you very much for your reminder. We have revised the description in the revised manuscript, which can be seen in the last sentence of Section 1.
Dimitrakellis [20] et al. pointed out that low surface energy antifouling coating with new nano antibacterial filler can achieve the effect of anti-biological attachment and the dual development of "active" antibacterial. In previous studies, the triazole ring structure widely used in biopharmaceutics can destroy biomass membrane [21-27], which is usually combined with fluorine to achieve a synergistic antibacterial mechanism. The fluorine's strong electronegativity and hydrogen-like mimic effect [28-32] harm bacteria's repro-duction and growth. Nevertheless, the two structures are seldom involved in marine antifouling. Therefore, we selected nano ZnO as the modified filler, modified its surface with triazole ring and fluoroaromatic hydrocarbon, to obtain a lightweight nano anti-bacterial filler as the "active" antibacterial. Furthermore, an antifouling coating was obtained after contained it in zinc acrylate resin. The low energy surface caused by fluorine can effectively prevent the adhesion of bacteria.
- [Experimental section] The remarks information for chemicals and testing equipments in this work should be given, for example ( Model, Name of supplier, City, Country).
Response4: We checked the experimental section and the source of each chemical reagent was added.
Nano zinc oxide (30 nm), triethylamine, 4-(trifluoromethyl) benzyl bromide, azide trimethylsilane (TMSA), tryptone, and yeast extract used in this study were purchased from Shanghai Macklin Biochemical Co., Ltd. Related Also 3-aminopropyl triethoxysilane. Copper sulfate pentahydrate was manufactured by Guangdong Sci Tech Co., Ltd. Sodium ascorbate was manufactured by Xiya Chemical Technology (Shandong) Co., Ltd. 3-Bromopropyne was obtained from Shanghai Aladdin Biochemical Technology Co., Ltd. Anhydrous ethanol was purchased from Xilong Science. Zinc acrylate resin was made in the laboratory. The fluorescent live and dead cell stain LIVE/DEAD BacLightTM Bacterial Viability Kit L7012 used in the study was purchased from the Thermo Fisher Scientific website. The conductivity of ultra-pure deionized water used must not exceed 0.2 μs/cm@25°C.
Staphylococcus aureus, Pseudoalteromonas sp., and Escherichia coli used in this ex-periment are from the State Key Laboratory of Hainan University.
- Found few spell mistakes in the current version. Avoid the typo errors and unnecessary space in the written manuscript.
Response5: Thank you for your suggestions. We have checked the paper again, and there are indeed many minor errors. We have made improvements in the revised v

Round 2
Reviewer 2 Report
Accept